# Rosmarinic Acid, a Component of Rosemary Tea, Induced the Cell Cycle Arrest and Apoptosis through Modulation of HDAC2 Expression in Prostate Cancer Cell Lines

**DOI:** 10.3390/nu10111784

**Published:** 2018-11-16

**Authors:** Yin-Gi Jang, Kyung-A Hwang, Kyung-Chul Choi

**Affiliations:** Laboratory of Biochemistry and Immunology, College of Veterinary Medicine, Chungbuk National University, Cheongju 28644, Chungbuk, Korea; mingue32@naver.com (Y.-G.J.); hka9400@naver.com (K.-A.H.)

**Keywords:** Rosmarinic acid, suberoylanilide hydroxamic acid (SAHA), histone deacetylase 2 (HDAC2), p53, cell cycle arrest and apoptosis

## Abstract

Rosmarinic acid (RA), a main phenolic compound contained in rosemary which is used as tea, oil, medicine and so on, has been known to present anti-inflammatory, anti-oxidant and anti-cancer effects. Histone deacetylases (HDACs) are enzymes that play important roles in gene expression by removing the acetyl group from histone. The aberrant expression of HDAC in human tumors is related with the onset of human cancer. Especially, HDAC2, which belongs to HDAC class I composed of HDAC 1, 2, 3 and 8, has been reported to be highly expressed in prostate cancer (PCa) where it downregulates the expression of p53, resulting in an inhibition of apoptosis. The purpose of this study is to investigate the effect of RA in comparison with suberoylanilide hydroxamic acid (SAHA), an HDAC inhibitor used as an anti-cancer agent, on survival and apoptosis of PCa cell lines, PC-3 and DU145, and the expression of HDAC. RA decreased the cell proliferation in cell viability assay, and inhibited the colony formation and tumor spheroid formation. Additionally, RA induced early- and late-stage apoptosis of PC-3 and DU145 cells in Annexin V assay and terminal deoxynucleotidyl transferase dUTP nick end labeling (TUNEL) assay, respectively. In western blot analysis, RA inhibited the expression of HDAC2, as SAHA did. Proliferating cell nuclear antigen (PCNA), cyclin D1 and cyclin E1 were downregulated by RA, whereas p21 was upregulated. In addition, RA modulated the protein expression of intrinsic mitochondrial apoptotic pathway-related genes, such as Bax, Bcl-2, caspase-3 and poly (ADP-ribose) polymerase 1 (*PARP-1*) (cleaved) via the upregulation of p53 derived from HDAC2 downregulation, leading to the increased apoptosis of PC-3 and DU145 cells. Taken together, treatment of RA to PCa cell lines inhibits the cell survival and induces cell apoptosis, and it can be used as a novel therapeutic agent toward PCa.

## 1. Introduction

Phenolic compounds found in tea are known to have anti-oxidant and anti-cancer effects [1]. Rosmarinic acid (RA) is a main phenolic compound in *Rosmarinus officinalis* L. (called rosemary) which is a common herb cultivated in many parts of the world and has been consumed as tea, oil, medicine and so on [2,3]. Previous studies on RA have reported its biological effects such as anti-inflammation [4], anti-diabetes [5] and especially anti-cancer effect against colorectal [6], gastric [7], ovarian [8], skin [9], liver [10] and breast cancer [11].

Prostate cancer (PCa) is the most leading type of cancer occurring in men and the second most common cause of cancer-related death worldwide [12]. Though chemotherapies, such as docetaxel, cabazitaxel, doxorubicin, mitoxantrone, and estramustine, have been used in treatment of PCa, these chemotherapies have some adverse side effects such as hair loss, nausea, vomiting, and fatigue [13]. Moreover, using the chemotherapeutic drugs in the long term allows aggressive PCa cells to experience mutations in the gene of beta-tubulin and activation of drug efflux pumps, leading to increased survival and the drug resistance [14,15,16]. 

Histone deacetylases (HDACs) are enzymes that play important roles in gene expression by removing the acetyl group from histone [17,18]. Based on their sequence homology, HDACs are classified into four classes such as class I (HDAC1, 2, 3 and 8), class II (HDAC4, 5, 6, 7, 9 and 10) and class IV (HDAC11) [19]. A number of studies related with HDACs have proved that the aberrant expression of HDAC is related with the onset of human cancer [20]. In diverse types of cancers, such as prostate [21], colorectal [22], breast [23], lung [24], liver [25] and gastric cancer [26], overexpression of HDACs is associated with a poor cancer prognosis and disease outcome, and can help to predict the tumor type and disease progression. Furthermore, the overexpression of HDACs has been highly associated with critical cancer-related phenomena such as the epigenetic repression of tumor suppressor genes like CDKN1A (encoding the cyclin-dependent kinase inhibitor p21) [27,28], and p53 resulting in its decreased transcriptional activity [29], and upregulation of oncogenes such as B-cell lymphoma-2 (BCL-2) [30]. Especially, high expression of HDAC2 which belongs to HDAC class I is observed in human epithelial cancer such as PCa, and downregulation of HDAC2 is related with growth arrest and apoptosis of PCa [21]. HDAC inhibitors, as a new class of anti-tumor agents, such as trichostatin A (TSA), suberoylanilide hydroxamic acid (SAHA), valproic acid, depsipeptide and sodium butyrate, are useful for the downregulation and inhibition of cancer growth [31,32].

The recent studies regarding the therapeutic properties of RA have shown that RA inhibits the cell proliferation via induction of the cell cycle arrest and apoptosis in colorectal cancer [6]. However, the detailed mechanisms underlying anti-cancer effects of RA on PCa has been not yet known. Therefore, based on the previous studies, we investigated the anti-PCa mechanisms of RA in association with its activity regulating HDAC2 expression. The abilities of RA to induce cell cycle arrest and apoptosis of PCa cells through HDAC inhibition were also identified in comparison with SAHA, a chemical inhibitor of HDAC2. By doing this, we examined the anti-PCa potential of RA as a novel phytochemical that can be substituted for the existing chemotherapeutic drugs including HDAC inhibitors.

## 2. Materials and Methods

### 2.1. Reagents and Chemicals

SAHA was purchased from Santa Cruz Biotechnology (Dallas, TX, USA) and RA (≥98% (HPLC)) was purchased from Sigma-Aldrich (St. Louis, MO, USA). All chemicals were dissolved in 100% dimethyl sulfoxide (DMSO, Junsei Chemical Co., Tokyo, Japan) which was used as a negative control (NC) and stocked at 10 ^−1^ M.

### 2.2. Cell Culture and Media

The human PCa cell lines, PC-3 and DU145, were purchased from the Korean Cell Line Bank (Seoul, Korea). Both cell lines were cultured using a medium (DMEM, HyClone Laboratories, Chicago, IL, USA) supplemented with 10% fetal bovine serum (FBS; RMBIO, Missoula, MT, USA), 1% penicillin G/streptomycin (Bio west, San Marcos, TX, USA), 1% HEPES (Gibco by Life Technologies, Gaithersburg, MD, USA) and 0.05% cell maxin (GenDEPOT, Katy, TX, USA) in cell culture dishes (SPL Life Science, Pocheon, Korea) at 37 °C in a humidified atmosphere containing 95% air and 5% CO_2_. Both cell lines were detached by using 0.05% Trypsin-EDTA (Gibco by Life Technologies, Gaithersburg, MD, USA).

### 2.3. Cell Viability Assay

Cell viability assay was performed to find the proper concentrations of SAHA and RA to inhibit viability of PCa cells. Both cell lines were seeded at 1 × 10^4^ cells per well in 96-well plates (SPL Life Science) in a humidified atmosphere of 5% CO_2_ at 37 °C. After the cells were incubated with medium for 24 h, the medium containing DMSO, SAHA (1, 2.5, 5, 10, 25 and 50 μM) and RA (25, 50, 100, 200, 250 and 300 μM) were treated for 48 h. Cell viability was determined using a EZ-Cytox cell viability assay kit (iTSBiO, Seoul, Korea). The medium in 96-well plates was gently removed, and then, EZ-Cytox reagent was dispensed to each well and incubated for 1 h under the standard cell culture condition. At last, 96-well plates were gently shaken and the absorbance at 450 nm of each well was measured by using an ELISA reader (Epoch, BioTek, Winooski, VT, USA).

### 2.4. Colonogenic Survival Assay

Colonogenic assay or colony formation assay is normally performed to estimate the in vitro cell survival activity based on the ability of single cells to grow into a colony [33,34,35]. Both cell lines were seeded at 5 × 10^3^ cells per well in 6-well plates (SPL Life Science) for 24 h, and then, the medium containing DMSO, SAHA (1 μM) and RA (200 μM) were added into plates and incubated for 2 weeks. Each medium was replaced every 4 days. After 2 weeks, all cells were fixed with 4% methanol-free formalin (Sigma-Aldrich) for 10 min and permeabilized with methanol (Sigma-Aldrich) for 10 min. After that, cells were stained with 0.5% crystal violet (hexamethylpararosaniline chloride; Sigma-Aldrich) for 10 min, and then washed with Dulbecco’s Phosphate-Buffered Saline (DPBS, WELGENE, Gyeongsan, Korea). The attached cells stained with crystal violet were pictured by using the camera (Samsung, Seoul, Korea) and counted by using the Image J program (National Institutes of Health, Bethesda, MD, USA).

### 2.5. Hanging Drop Assay Detecting for Tumor Spheroid Formation

Hanging drop assay allows for the formation of spheroid shaped tumors by self-assembly of tumor colonies, which can be used to evaluate chemotherapeutic drugs in a biological environment closer to in vivo models [36,37]. Both cell lines in media containing DMSO, SAHA (1 μM) and RA (200 μM) were seeded on petri dish covers (SPL Life Science) at 3 × 10^3^ cells in 25 μL each medium by using a multi-pipette to form spheroids. After a week, the droplets were gently gathered in 6-well plates (SPL Life Science) and photographed by using the IX-73 inverted microscope (Olympus, Tokyo, Japan). The size of tumor spheroids in each droplet was measured by using the Image J program (National Institutes of Health).

### 2.6. Annexin V Assay

Firstly, to confirm whether SAHA and RA are effective to induce early and late apoptosis, Annexin V assay was performed following the protocol of Alexa Fluor 488 annexin V/Dead Cell Apoptosis kit (Invitrogen, Carlsbad, CA, USA). Concisely, both cell lines were seeded in cell culture dishes at 7 × 10^5^ cells/10 mL for 24 h and cultured in medium containing DMSO, SAHA (1 μM) and RA (200 μM) for 48 h. Then, cells of each group were detached by trypsin and centrifuged. The cells were placed in 1× annexin-binding buffer containing Alexa Fluor 488 annexin V (Alexa), propidium iodide (PI) and Alexa + PI under dark condition at room temperature for 15 min. The apoptotic cells at each stage were analyzed by using the flow cytometry (Sony SH800 Cell sorter, Tokyo, Japan).

### 2.7. TUNEL Assay

To examine whether SAHA and RA affect DNA fragmentation occurring at a stage of late apoptosis, TUNEL assay was performed by using a DeadEnd™ fluorometric terminal deoxynucleotidyl transferase (TdT)-mediated deoxyuridine triphosphate (dUTP) nick-end labeling (TUNEL) system (Promega, Madison, WI, USA) following the manufacturer instructions. Both cell lines were seeded at 3 × 10^5^ cells per well in 24-well plates with media and incubated for 24 h. Next day, the medium containing DMSO, SAHA (1 μM) and RA (200 μM) were added into plates. After treatment for 48 h, the cells were fixed with 4% methanol-free formalin (Sigma-Aldrich) for 25 min at 4 °C, and washed by DPBS for 5 min. Permeabilization was done by using a lysis buffer (1% Triton X-100 in 1% sodium citrate) for 5 min, and then treated with 50 μL TdT enzyme buffers composed of equilibration buffer, nucleotide mix and rTdT enzyme which were bound to DNA strand breaks. Finally, labeled strand breaks were confirmed through the attachment of fluorescein isothiocyanate-5-dUTP. After staining apoptotic cells with TUNEL assay kit, every well was counterstained with a 4, 6-diamidino-2-phenylindole (DAPI; Invitrogen) and observed by using a fluorescence microscope (IX-73 Inverted Microscopy, Olympus). The apoptotic activity of SAHA and RA on PC3-3 and DU145 cells was quantified and analyzed using the Cell Sens Dimension software 1.13 (Build 13479, Olympus).

### 2.8. Western Blot Analysis

To determine the protein expression of HDAC2 and of genes involved in cell cycle and apoptosis regulation, western blot analysis was performed. Both cell lines were seeded at 1 × 10^6^ in cell culture dishes for 24 h and treated with DMSO, SAHA (1 μM) and RA (200 μM) for 48 h. Then, whole cell lysates were prepared by RIPA buffer which was composed of 50 mM Tris, 150 mM NaCl, 1% Triton X-100 (Sigma-Aldrich), 0.5% deoxycholic acid (Sigma-Aldrich), and 0.1% SDS and protease inhibitor (GenDEPOT, Katy, TX, USA). Total proteins that were extracted from the cell lysates were quantified using bicinchoninic acid (BCA; Sigma-Aldrich). Protein mixtures (protein, distilled water (DW) and dye) were then loaded and separated in SDS-polyacrylamide gel electrophoresis (SDS-PAGE), and then transferred to polyvinylidene difluoride (PVDF) membranes (Bio-Rad, Hercules, CA, USA). Finally, the membranes were blocked in 5% skim milk (Blotting-Grade Blocker; Bio-Rad) for an hour at 4 °C on a shaker. The membranes were incubated overnight at 4 °C in BCA containing primary antibodies shown in Table 1. Washing steps were performed by using 1X Tris-buffered saline Tween (TBS-T; 50 mM Tris and 150 mM NaCl with 0.1% Tween 20 (GenDEPOT), pH: 7.6). Primary antibodies bound to membrane were detected with horse radish peroxidase (HRP)-conjugated anti-mouse IgG or anti-rabbit IgG (1:2000, Thermo Scientific, Waltham, MA, USA) incubated for 2 h at room temperature. Targeted bands were detected by using the Ez west-Lumi plus (ATTO, Tokyo, Japan) and Lumino graph II (ATTO). The expression levels were quantified and normalized by using the CS Analyzer4 (ATTO).

### 2.9. Statistical Analysis

All experiments were conducted at least three times, and all data were statistically analyzed with the Graph-pad Prism software (Graph-pad software Inc, San Diego, CA, USA). Data were expressed as the means ± standard deviation (SD) and analyzed by one-way analysis of variance (ANOVA) followed by Dunnett’s multiple comparison test. *p*-Values of <0.05 indicates statistical difference versus control.

## 3. Results

### 3.1. RA Decreased the Viability of PCa Cell Lines in a Dose-Dependent Manner

To select the inhibition concentrations of SAHA and RA on PCa cell viability, the cell viability assay was done in PC-3 and DU145 cells. The culture medium containing 0.2% DMSO was used as NC because the same concentration of DMSO was employed as a vehicle to dissolve SAHA or RA. SAHA and RA were administered to both cell lines for 48 h and the cell viability of PCa cells was confirmed by Water Soluble Tetrazolium (WST) salt assay. As a result, SAHA significantly decreased the cell viability of both cell lines in a dose-dependent manner in the concentration range tested (1–50 μM) (Figure 1A). For RA, it slightly decreased cell viability of both cell lines at 25, 50, and 100 μM, but significantly inhibited cell viability (>50%) at the concentrations of higher than 200 μM (Figure 1B). Based on these results, 1 μM SAHA and 200 μM RA were selected for further experiments. 

### 3.2. RA Inhibited the Formation of Colonies of PCa Cell Lines

The inhibition ability of RA against PCa cell survival was confirmed by colony formation assay. After treatment with NC (DMSO), SAHA and RA for 2 weeks, colonies formed by each cell line were stained and quantified. The results showed a high number of colonies in NC, but only few colonies in SAHA and RA treatments as the almost cells were dead (Figure 2A). According to these results, SAHA and RA significantly inhibited the formation of colonies. It was shown that RA considerably decreased the survival activity of PCa cell lines, as SAHA did. 

In addition to colony formation assay, hanging drop assay was conducted to examine the effects of SAHA and RA on tumor spheroid formation of PCa cells. The results showed that the larger tumor spheroids were formed in NC, while small-size spheroids were detected in SAHA and RA treatments. The inhibition effect of SAHA and RA on tumor spheroid formation was more conspicuous for PC-3 than for DU145 (Figure 2C,D). Referring to the results of tumor spheroid formation assay, RA effectively interrupted 3 dimensional (3D) tumor formation of PCa cells, as SAHA did.

### 3.3. RA Induced Apoptosis in PCa Cell Lines

To check which phase of apoptosis is mainly induced by RA, Annexin V assay was performed after treatment with NC, SAHA and RA. In this assay, the protein Annexin V detects early and late stages of apoptosis. The cells undergoing apoptosis or not are detected by flow cytometry and divided into 4 groups (Q1, Q2, Q3 and Q4), which indicate necrosis (Annexin V-negative/PI-positive), late apoptosis (Annexin V-positive/PI-positive), live state (Annexin V-negative/PI-negative) and early apoptosis (Annexin V-positive/PI-negative). The results showed that RA mainly increased the number of cells in late apoptosis and necrosis, while SAHA mainly increased the number of cells in early apoptosis in DU-145 cells. For PC-3 cells, RA increased the number of early apoptotic cells, and SAHA increased the number of late apoptotic cells compared to NC (Figure 3).

### 3.4. RA Induced the DNA Fragmentation in PCa Cell Lines

To investigate whether RA is effective in DNA fragmentation related with induction of late apoptosis, compared to SAHA in PC-3 and DU145 cell lines, TUNEL assay was performed after treatment with NC, SAHA and RA. Cell nuclei were stained in blue with DAPI, and DNA fragmentation occurring in apoptotic cells was detected as green fluorescence by TUNEL. As shown in Figure 4, apoptotic cells were rarely detected in NC, but in case of SAHA and RA treatment, the apoptotic cells observed by DNA fragmentation were significantly increased in comparison with NC in both cell lines (Figure 4).

### 3.5. RA Downregulated the Expression of HDAC2 and p53 in PCa Cell Lines

To confirm the effects of RA on HDAC2 and p53 expression at the protein level, western blot analysis was performed. As shown in Figure 5, RA significantly reduced the protein expression of HDAC2, similarly to SAHA in PC-3 cells, by half as much as SAHA in DU145 cells. In case of p53, RA significantly increased its protein expression in both cell lines, and SAHA significantly increased the p53 expression only in PC-3 cells, but decreased its expression in DU145 cells as compared to NC. These data suggested that RA effectively reduced the protein expression of HDAC2 and p53 in both PCa cell lines. For SAHA, it displayed a similar result to that of RA in PC-3 cells, but a significant decrease of p53 in DU145, although it apparently decreased the HDAC2 protein expression (Figure 5).

### 3.6. RA Regulated the Expression of Cell Cycle-Related Genes in PCa Cell Lines

To further confirm the mechanisms related with modulation of HDAC2 and p53 by RA treatment, western blot analysis pertaining to the expression of cell cycle-related genes was done. Based on these results, RA significantly increased the expression of p21 only in PC-3 cells, but there was no change in DU145 cells. Despite this situation, RA significantly decreased the expression of cell cycle-related genes, such as *PCNA*, cyclin D1 and cyclin E1 in both cell lines, as SAHA did. These data suggested that RA inhibited the cell proliferation via the induction of cell cycle arrest, similar to SAHA (Figure 6).

### 3.7. RA Regulated the Expression of Apoptosis-Related Genes in PCa Cell Lines

The protein expression of genes related with apoptosis was confirmed by conducting the western blot analysis. The expression of Bcl-2-associated X (Bax) protein was greatly upregulated by SAHA and RA in PC-3 cells, but only upregulated by RA in DU145 cells, likewise the expression of p53. The expression of Bcl-2 was significantly downregulated by treatment with SAHA and RA in both cell lines. The expression of Caspase-3 was upregulated by SAHA and RA in both cell lines, but PARP-1was cleaved in the cells treated with RA only. These results indicated that RA can promote apoptosis of PCa cells by regulating the protein expression of apoptosis-related genes. However, SAHA did not stimulate the expression of apoptotic genes, which RA did (Figure 7).

## 4. Discussion

RA, which is easily taken from tea and vegetables, is a dietary phytochemical that has diverse bioactivities, such as anti-bacterial, anti-oxidative, and anti-inflammatory effects [38]. Recently, RA has been getting attention because it is known to have anticancer property with relatively low toxicity compared with conventional chemotherapies that have severe side effects and drug resistance [14,15,16]. However, its anti-cancer effect and related mechanisms on PCa have not yet been known, despite the fact that the chemopreventive effects of several phytochemicals including genistein, resveratrol, kaempferol and catechin on PCa have been identified [39,40,41,42]. In this study, we tried to discover the anti-cancer effect of RA on PCa, along with its underlying mechanism associated with HDAC2 inhibition and apoptosis induction. 

Firstly, RA and SAHA were found to significantly inhibit the cell viability of PC-3 and DU145 cell lines (Figure 1). RA decreased the cell viability of both PCa cell lines by about 50% at 200 μM and sharply decreased the cell viability at higher concentrations than that. For SAHA, it inhibited the cell viability of both PCa cell lines in a dose-dependent manner in the concentration range of 1–50 μM and significantly decreased the cell viability even at 1 μM (a 40% decrease for PC-3 and a 60% decrease for DU145). Based on these results, the concentrations of RA and SAHA for the next experiments were selected as 200 μM and 1 μM, respectively. It was also found that RA and SAHA regulated the protein expression of cell cycle-related genes in the direction of cell cycle arrest at these concentrations (Figure 6). 

In addition, the effects of RA and SAHA on the colony formation and tumor spheroid formation were displayed to correspond with the results of cell viability; RA and SAHA remarkably suppressed the colony and tumor spheroid formation of both PCa cell lines at these concentrations (Figure 2). Colony formation capacity of cells reflects their survival and proliferation abilities, and spheroid formation is associated with the ability of cells that can grow in all directions by interacting with themselves or their surroundings, as in the in vivo culture [43,44]. Therefore, these findings indicate that RA can inhibit the viability and proliferation of PCa cells as well as blocking the formation of tumor spheroids that resemble the condition of in vivo tumor tissue, like SAHA.

Secondly, apoptotic activity of RA and SAHA on PCa cells were identified in Annexin V assay and flow cytometry; RA mainly induced the late apoptosis and necrosis in PC-3 and DU145 cell lines, unlike SAHA which was effective in induction of early apoptosis in both cell lines (Figure 3). However, in PC-3 cells, RA also induced early apoptosis, and SAHA also induced late apoptosis. Meanwhile, RA and SAHA significantly induced DNA fragmentation in both PCa cell lines, which was identified by TUNEL assay (Figure 4). Although there were differences in the individual effect of RA and SAHA on the protein expression of apoptosis-related genes in both PCa cell lines, RA and SAHA were found to induce apoptosis of PCa cells by upregulating caspase-3 and downregulating Bcl-2 (Figure 7). For RA, it additionally increased the protein expression of Bax and cleaved form of PARP-1. Pro-apoptotic protein Bax and anti-apoptotic protein Bcl-2 family members bound to the mitochondrial membrane affect the intrinsic mitochondrial pathway of apoptosis by regulating efflux of cytochrome c from mitochondria to cytoplasm and subsequent activation of caspases such as caspase-3, -6, and -7 that are cysteine-aspartic proteases playing important roles in apoptosis [45]. PARP-1 is a poly-(ADP-ribosylating) enzyme necessary for DNA repair processes. During apoptosis, it is cleaved by caspase-3 or -7 and becomes inactive in recovering DNA damage. Thus, cleaved form of PARP-1 is considered as a remarkable marker of apoptosis [46]. As a result, RA was revealed to effectively induce apoptosis of PCa cells by influencing an intrinsic mitochondrial apoptotic pathway. 

As an inhibitor of HDAC1 and HDAC2, SAHA, also known as Vorinostat, has a broad spectrum of epigenetic activities through the inhibition of histone acetylation and has been used to treat several diseases including glioblastoma [47] and non-small-cell lung carcinoma [48]. In association with tumor, HDAC1 and HDAC2 enhance the tumorigenesis by decreasing the transcriptional activity of tumor suppressor like p53 through histone deacetylation. In addition, deacetylation of lysine residues of p53 by HDACs leads to ubiquitinylation and proteolysis of p53 by the proteasome. Therefore, HDACs downregulate p53 activity, which cannot block anymore the cell cycle progression and trigger apoptosis [49]. On the other hand, SAHA can have anti-cancer efficacies by inhibiting HDACs and upregulating p53 [50]. In the present study, SAHA was found to effectively suppress the protein expression of HDAC2 in both PCa cell lines, and the protein expression of p53 was dramatically increased as a result of HDAC2 inhibition in PC-3 cells. However, p53 was not upregulated in DU145 cells beyond expectations (Figure 5). Nevertheless, the growth inhibition effects or apoptotic activities of SAHA on DU145 cells were found on PC3 cells, indicating that SAHA may induce apoptosis of DU145 cells through p53-independent pathway. In the previous study on U937 human leukemia cells, SAHA was revealed to induce apoptosis through the pathway independent of p53 [51]. The more detailed mode of apoptotic action of SAHA in DU145 cells needs to be elucidated. In comparison with SAHA, RA inhibited the protein expression of HDAC2 and increased the protein expression of p53 in both PCa cell lines, which was more noticeable in DU145 cells than in PC-3 cells. 

p53, as a tumor suppressor protein, has diverse anti-cancer effects such as cell cycle arrest, apoptosis induction, and inhibition of angiogenesis [52]. Cell cycle arrest mediated by p53 is known to be partially achieved through activating p21, which hinders G1/S transition in the cell cycle as an inhibitor of CDK complexes [53,54]. p53 also mediates apoptosis through upregulating the expression of Bax, a pro-apoptotic gene, which prevents the activity of Bcl-2, an anti-apoptotic gene [55]. In the present study, it was confirmed that cell cycle arrest and apoptosis of PCa cells were achieved in parallel by p53, which was upregulated by RA and SAHA as a result of HDAC2 inhibition. 

Among numerous anticancer mechanisms of plant-derived phytochemicals, it is known that they are related with apoptotic cell death that is mediated by p53-dependent or -independent pathways. A recent review emphasized the cell death mechanisms of diverse plant-derived anti-cancer polyphenolics, alkaloids, terpenoids, and so on [56]. The present study is considered to firstly identify the cell death mechanism of RA on PCa cells that is relevant to p53-induced apoptosis caused by HDAC2 inhibition.

## 5. Conclusions

RA, as a dietary phenolic compound ingested from tea, was displayed to have anti-PCa activities by inhibiting viability, colony formation, and spheroid formation of PCa cells via HDAC2 inhibition and the consequential p53-mediated cell cycle arrest and apoptosis Therefore, RA would be used as a novel phytomedicine to act as an HDAC inhibitor targeted to PCa, with the anticipation to decrease the adverse side effects of the existing chemotherapeutical agents. To do this, further studies are necessary to identify eventual cytotoxic effects of RA on normal cells. In addition, supraphysiologic doses of RA should be attained to produce anti-cancer effects on PCa with the formulation containing RA as a dietary additive or a drug containing its effective therapeutic concentration.

## Figures and Tables

**Figure 1 nutrients-10-01784-f001:**
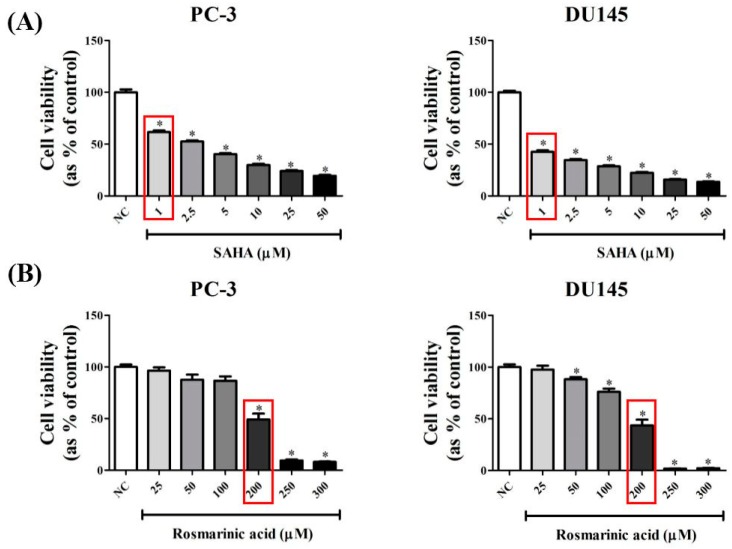
Effects of suberoylanilide hydroxamic acid (SAHA) and Rosmarinic acid (RA) on cell viability in PC-3 and DU145 cell lines. After PC-3 and DU145, cell lines were seeded at 1 × 10^4^ cells per well in 96-well plates and treated with media containing negative control (NC; DMSO), SAHA (1, 2.5, 5, 10, 25, 50 μM) and RA (25, 50, 100, 200, 250, 300 μM), cell viability of each cell line was evaluated. The data showed (**A**) the effect of SAHA and (**B**) the effect of RA on cell viability of both cell lines. The results are expressed as means ± standard deviation (SD). * *p* < 0.05: a significant difference versus control.

**Figure 2 nutrients-10-01784-f002:**
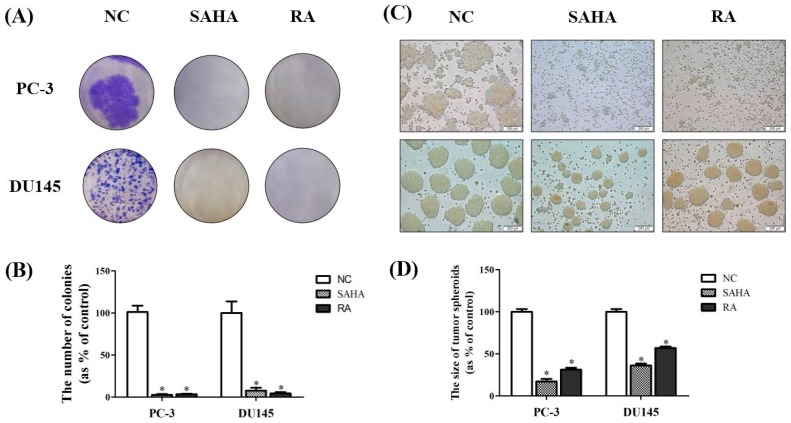
Effects of SAHA and RA on formation of colonies and tumor spheroids in PC-3 and DU145 cell lines. (**A**) After both cell lines were seeded at 5 × 10^3^ cells per well in 6-well plates and treated with media containing NC (DMSO), SAHA (1 μM) and RA (200 μM), colonogenic assay was performed to measure the colony formation. (**B**) The number of colonies was quantified by using the Image J program. (**C**) Both cell lines in media containing NC (DMSO), SAHA (1 μM) and RA (200 μM) were seeded on petri dish covers at 3 × 10^3^ cells in 25 μL. (**D**) The size of tumor spheroids was measured by using the Image J program. The results are expressed as means ± SD. * *p* < 0.05: a significant difference versus control.

**Figure 3 nutrients-10-01784-f003:**
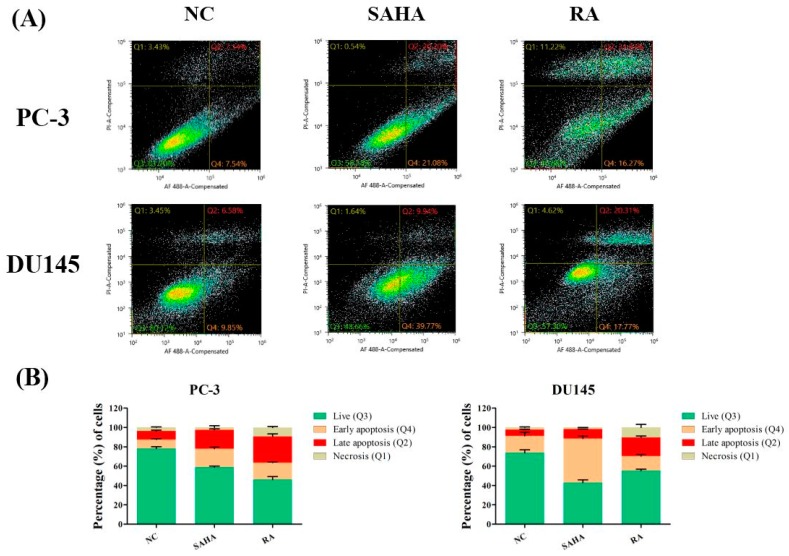
Effects of SAHA and RA on the apoptotic events in PC-3 and DU145 cell lines. (**A**) After both cell lines were seeded at 7 × 10^5^ cells in cell culture dishes and treated with media containing NC (DMSO), SAHA (1 μM) and RA (200 μM), Annexin V assay was conducted. (**B**) The apoptotic cells at each stage were analyzed and quantified by using the flow cytometry. Q1, Q2, Q3 and Q4 indicate necrosis, late apoptosis, live and early apoptosis, respectively.

**Figure 4 nutrients-10-01784-f004:**
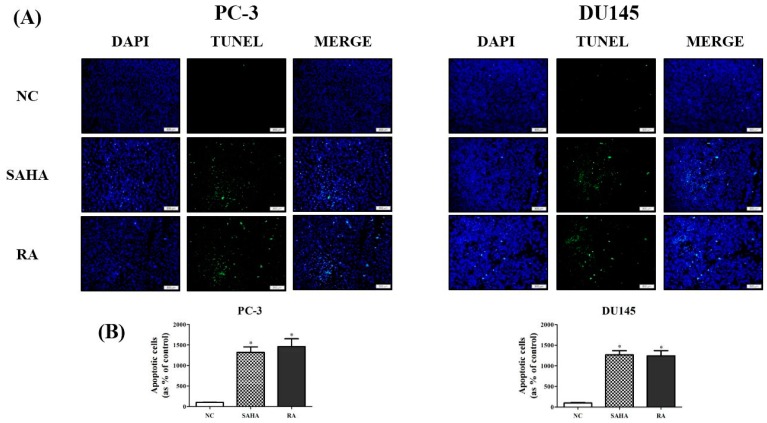
Effects of SAHA and RA on DNA fragmentation in PC-3 and DU145 cell lines. After both cell lines were seeded at 3 × 10^5^ cells per well in 24-well plates and treated with media containing NC (DMSO), SAHA (1 μM) and RA (200 μM), terminal deoxynucleotidyl transferase dUTP nick end labeling (TUNEL) assay was performed. (**A**) The data indicated that nuclei were stained with 4′,6-diamidino-2-phenylindole (DAPI), and DNA fragmentation was stained with TUNEL. DAPI and TUNEL were merged to observe apoptotic nuclei. (**B**) The apoptotic cells presenting DNA fragmentation were quantified separately. The results are expressed as means ± SD. * *p* < 0.05: a significant difference versus control. MERGE means the combined picture of DAPI and TUNEL pictures.

**Figure 5 nutrients-10-01784-f005:**
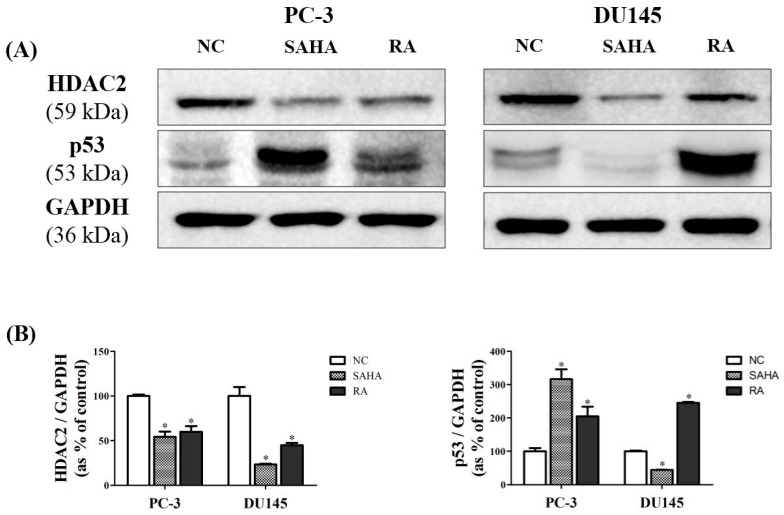
Effects of SAHA and RA on expression of histone deacetylase 2 (HDAC2) and p53 in PC-3 and DU145 cell lines. After both cell lines were seeded at 1 × 10^6^ in cell culture dishes and treated with media containing NC (DMSO), SAHA (1 μM) and RA (200 μM), the western blot analysis was performed. (**A**) The expressions of HDAC2 and p53 at the protein level were confirmed by western blot analysis. (**B**) The expression levels of HDAC2 and p53 were quantified and normalized to glyceraldehyde-3-phosphate dehydrogenase (GAPDH). The results are expressed as means ± SD. * *p* < 0.05: a significant difference versus control.

**Figure 6 nutrients-10-01784-f006:**
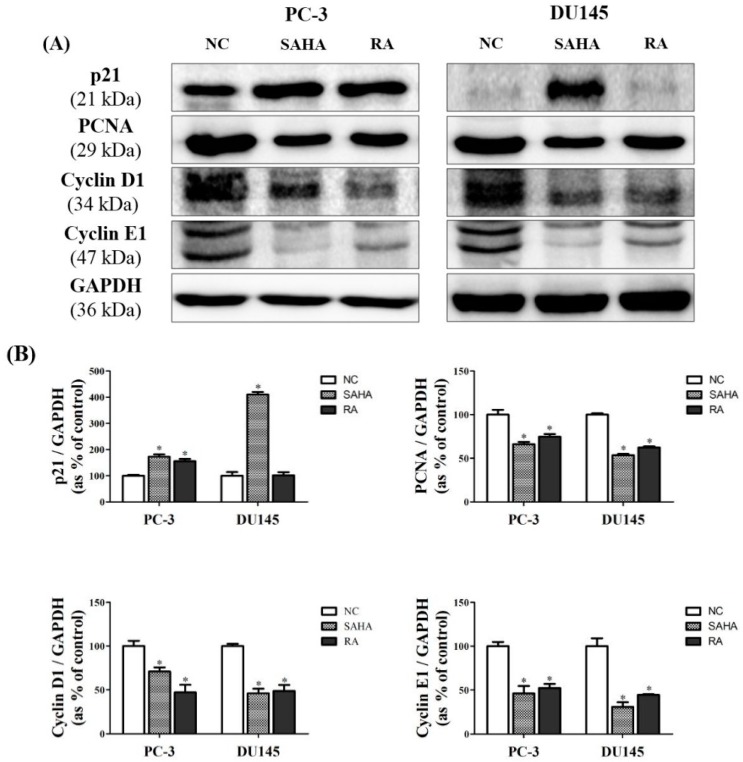
Effects of SAHA and RA on expression of cell cycle related genes in PC-3 and DU145 cell lines. After both cell lines were seeded at 1 × 10^6^ in cell culture dishes and treated with media containing NC (DMSO), SAHA (1 μM) and RA (200 μM), the western blot analysis was performed. (**A**) The expressions of p21, proliferating cell nuclear antigen (PCNA), cyclin D1 and cyclin E1 at the protein level were confirmed by western blot analysis. (**B**) The expression levels of each gene were quantified and normalized to GAPDH. The results are expressed as means ± SD. * *p* < 0.05: a significant difference versus control.

**Figure 7 nutrients-10-01784-f007:**
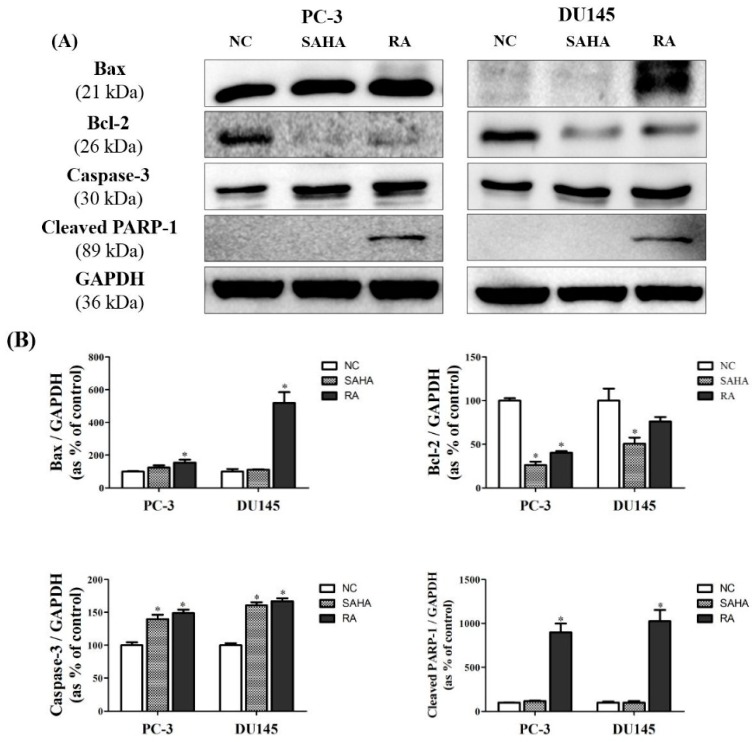
Effects of SAHA and RA on expression of apoptosis related genes in PC-3 and DU145 cell lines. After both cell lines were seeded at 1 × 10^6^ in cell culture dishes and treated with media containing NC (DMSO), SAHA (1 μM) and RA (200 μM), the western blot analysis was performed. (**A**) The expressions of Bax, Bcl-2, caspase-3 and poly [ADP-ribose] polymerase 1 (PARP-1) (cleaved) at the protein level were confirmed by western blot analysis. (**B**) The expression levels of each gene were quantified and normalized to GAPDH. The results are expressed as means ± SD. * *p* < 0.05: a significant difference versus control.

**Table 1 nutrients-10-01784-t001:** Information of antibodies used in this study.

Antibody Name	Company	Description	Dilution
HDAC2	Santa cruz (Dallas, TX, USA)	Mouse monoclonal	1:1000
p53			1:200
PARP-1 (cleaved)			1:200
Caspase-3	Flarebio (College Park, Maryland)	Rabbit polyclonal	1:1000
PCNA	Abcam (Cambridge, UK)	Mouse monoclonal	1:10,000
Cyclin D1		Mouse monoclonal	1:2000
Cyclin E1		Rabbit polyclonal	1:2000
Bax		Mouse monoclonal	1:1000
Bcl-2		Mouse monoclonal	1:1000
GAPDH		Mouse monoclonal	1:12,000

HDAC2: histone deacetylase 2; PCNA: proliferating cell nuclear antigen; PARP-1: poly (ADP-ribose) polymerase 1; GAPDH: glyceraldehyde-3-phosphate dehydrogenase.

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
