# Peer review of "Rosmarinic Acid, a Component of Rosemary Tea, Induced the Cell Cycle Arrest and Apoptosis through Modulation of HDAC2 Expression in Prostate Cancer Cell Lines"

_nutrients, 2018, doi:10.3390/nu10111784_

Round 1
Reviewer 1 Report
1. Rosmarinic acid has anticancer properties against different cancer types, such as gastric cancer, colorectal cancer, liver cancer, breast cancer, ovarian cancer, etc. This information should be included in Introduction section.
2. In this in vitro study, the dose of rosmarinic acid starts from 25 µmoml/L and the highest concentration is 300 µmol/L. The concentration to achieve >50% cell viability inhibition is more than 200 µmol/L.
In animal study: The level of plasma total rosmarinic acid (free and conjugate forms) reached a maximum concentration of 4.63µmol/L after orally administration to the rats at a dose of 50mg/kg body weight (Life Sci. 2004, 75:165-178, PMID 15120569).
In human study: The level of serum total rosmarinic acid reached a maximum concentration of 162.20 nmol/L after orally administration to healthy individuals at a dose of 500mg rosmarinic acid (PLoS One. 2015, 10:e0126422, PMID: 25978046).
Based on the literature report, it seems that the authors used a supraphysiological dose of rosmarinic acid to achieve favorable effects. How could the findings in this study be translated into clinical practice?
3. The rosmarinic acid concentration should be indicated in tumor spheroid formation assay, Annexin V assay, and TUNEL assay (page 4).
4. There are at least 12 isoforms of p53 proteins. Some p53 proteins act as tumor suppressor, and some of which have tumor-promoting properties. Which isoforms of p53 protein was detected in this study?
Author Response
1. Rosmarinic acid has anticancer properties against different cancer types, such as gastric cancer, colorectal cancer, liver cancer, breast cancer, ovarian cancer, etc. This information should be included in Introduction section.
Ă The information about anticancer effects of Rosmarinic acid (RA) on diverse cancers has been added in Introduction section, as requested.
2. In this in vitro study, the dose of rosmarinic acid starts from 25 µmoml/L and the highest concentration is 300 µmol/L. The concentration to achieve >50% cell viability inhibition is more than 200 µmol/L.
In animal study: The level of plasma total rosmarinic acid (free and conjugate forms) reached a maximum concentration of 4.63µmol/L after orally administration to the rats at a dose of 50mg/kg body weight (Life Sci. 2004, 75:165-178, PMID 15120569).
In human study: The level of serum total rosmarinic acid reached a maximum concentration of 162.20 nmol/L after orally administration to healthy individuals at a dose of 500mg rosmarinic acid (PLoS One. 2015, 10:e0126422, PMID: 25978046).
Based on the literature report, it seems that the authors used a supraphysiological dose of rosmarinic acid to achieve favorable effects. How could the findings in this study be translated into clinical practice?
à As pointed out by the reviewer, the concentration of RA tested in this study (200 μM) was a supraphysiological level. In order to identify the anti-prostate cancer effect and the underlying mechanism of RA in vitro, we used the high dose of RA, as in other studies (200 μM RA used in “Rosmarinic Acid Activates AMPK to Inhibit Metastasis of Colorectal Cancer”. Front Pharmac ol 9:68. 2018/40 μM & 160 μM RA used in “Anticancer effects of Rosmarinic acid in OVCAR-3 ovarian cancer cells are mediated via induction of apoptosis, suppression of cell migration and modulation of lncRNA MALAT-1 expression”. J BUON. 2018 May-Jun;23(3):763-768). In fact, it is not possible to translate the findings of this study into clinical practice because we identified antiprostate cancer effect of RA in the in vitro cellular model only, not in the in vivo animal model. Therefore, we should determine the effective in vivo dose of RA in further studies. For instance, in case that higher concentrations of RA than its physiological concentration are needed to target prostate cancer, it seems to be necessary to develop RA into anti-prostate cancer medicine and use it actively for medical purposes, not only depending on its dietary ingestion from foods including RA. In conclusion, it can be said that this study provides a possibility of RA for antiprostate cancer therapy and needs more follow-up researches.
3. The rosmarinic acid concentration should be indicated in tumor spheroid formation assay, Annexin V assay, and TUNEL assay (page 4).
Ă The concentrations of SAHA and RA used in those assays have been indicated, as requested.
4. There are at least 12 isoforms of p53 proteins. Some p53 proteins act as tumor suppressor, and some of which have tumor-promoting properties. Which isoforms of p53 protein was detected in this study?
Ă We focused on the DO-1 form of p53, which is known as a tumor suppressor to promote growth arrest and apoptosis, and purchased its antibody [sc-126 from Santa Cruz Biotechnology] to analyze its protein expression in western blot assay.

Reviewer 2 Report
This is a well-written manuscript with reasonable science and interesting results.
My major concern is the organization of the manuscript.
I have some concerns about:
a) Lines 45-47. The sentence that begins with “Plus (..)” must be replaced for another one that includes the resistance mechanisms of anti-cancer drugs in general and not the particular case of docetaxel.
b) Lines 48-64. This information must be replaced the goal of the study is prostate cancer and not cancer in general.
c) Introduction: please describe the novelty of this study in comparison with previous studies.
d) Line 75. Introduce the % of purity of Rosmarinic acid (RA).
e) Results. Line 172. There are a lot of information that must be replaced to material and methods and not in the results section.
d) Line 226. Please explain better the results and compare the difference between the results with RA and SAHA treatment.
e) Line 298. Add cathechins.
f) Line 306. The reasons why the authors chose the 200 and 1 micromolar are not explained. Is that possible to reach 200 micromolar with RA in vivo after how much amount of tea??Probably with a lot (L?)..this must be discussed as well the bioavailability of RA.
g) Line 350. This information about p53 is basic and not relevant for the discussion of the results.
h) There is no conclusion section and some of the information of the results discussion, for instance the last paragraph is clearly a conclusion.
Please correct:
- Line 37: italic for the name of the specie.
Author Response
This is a well-written manuscript with reasonable science and interesting results.
My major concern is the organization of the manuscript. I have some concerns about:
a) Lines 45-47. The sentence that begins with “Plus (..)” must be replaced for another one that includes the resistance mechanisms of anti-cancer drugs in general and not the particular case of docetaxel.
Ă Brief information about the resistance mechanisms of anti-cancer drugs in general has been added, as requested.
b) Lines 48-64. This information must be replaced the goal of the study is prostate cancer and not cancer in general.
Ă We just briefly explained that HDAC is expressed in not only prostate cancer, but also various cancers to emphasis the important role of HDAC in cancers. In addition, pro-cancer activity of HDAC in prostate cancer was also described. Therefore, it is considered that this information does not weaken the argument of this study.
c) Introduction: please describe the novelty of this study in comparison with previous studies.
Ă Comparing with the previous studies reporting anti-cancer effects of RA on other cancer types, there has been not yet a report that dealt with in detail anti-prostate cancer effect and mechanism of RA. In this respect, we considered that this study has a novelty and added the fact like this in Introduction section, as requested.
d) Line 75. Introduce the % of purity of Rosmarinic acid (RA).
Ă The purity of RA has been described, as requested.
e) Results. Line 172. There are a lot of information that must be replaced to material and methods and not in the results section.
Ă As requested, the experimental information has been deleted in the results section, as requested, since it was already included in Material and Methods.
d) Line 226. Please explain better the results and compare the difference between the results with RA and SAHA treatment.
Ă The difference in results of RA and SAHA treatment has been explained in Results 3.3, paragraph, line 215-222.
e) Line 298. Add cathechins.
Ă Catechin has been added in line 298 and we also added the reference related with catechin.
f) Line 306. The reasons why the authors chose the 200 and 1 micromolar are not explained. Is that possible to reach 200 micromolar with RA in vivo after how much amount of tea??Probably with a lot (L?)..this must be discussed as well the bioavailability of RA.
Ă The reasons to select the concentrations of RA and SAHA was explained in Result and Discussion section. In fact, it is not possible to reach 200 ÎĽM of RA in vivo from ingestion of tea. If RA is developed to a kind of anticancer medicine, it can be purposely administered to the body at supraphysiological levels, not via its dietary ingestion from foods.
g) Line 350. This information about p53 is basic and not relevant for the discussion of the results.
Ă p53 as a typical tumor suppressor protein is known to have diverse anticancer modes of action. In the present study, we found that RA has antiprostate cancer effects by upregulating p53 as a result of HDAC2 inhibition. It was also confirmed that p53 plays an important role in induction of cell cycle arrest and apoptosis of prostate cancer cells, as shown in Fig. 8, a summary figure. In this respect, p53 is considered a major intermediate protein to mediate the antiprostate cancer activity of RA. Therefore, we added the information about p53 which is related with its role in induction of cell cycle arrest and apoptosis of prostate cancer cells by subsequently upregulating p21 and BAX, respectively. This information is considered to be closely associated with the results of this study.
h) There is no conclusion section and some of the information of the results discussion, for instance the last paragraph is clearly a conclusion.
Ă The conclusion section has been made, as requested.
Please correct:
- Line 37: italic for the name of the specie.
Ă The names of species have been written in italic, as requested.

Reviewer 3 Report
This manuscript reports on the rosmarinic acid (RA) anticancer effects exerted on prostate cancer cell lines (PC-3 and DU145) through cell cycle arrest and apoptosis mechanisms. Although these anticancer effects of RA are known, the associated molecular mechanism are not yet elucidated. Therefore, the present manuscript is original and presents a real interest for the scientific community. The experiments are elegant and scientifically correct, the results are fairly presented and the discussion comprehensive and well documented. However, the manuscript needs major revision mainly from the viewpoint of English language and style. Several suggestions are proposed below to the authors to improve their manuscript.
Scientific comments
- L46 – L47 (“microtubule mutations”): Microtubules are components of cytoskeleton that is composed of proteins alpha- and beta-tubulin. The gene of beta-tubulin is affected by mutation, not the microtubule, although the function of this last one is disturbed by this mutation. Please correct accordingly.
- L54: The following expression does not seem appropriate in the context of cancer evolution: “is responsible for the decrease in disease-free and survival”. The authors may mean “is associated with a poor cancer prognosis and disease outcome”. Please correct accordingly.
- L56 – 57: The deacetylation of p53 on its lysine residues is also an epigenetic regulation that leads to p53 ubiquitinylation and proteolytic degradation. Thus, its activity as a transcription factor is blocked. The sentence is proposed to be rephrased as explained below within the “minor corrections” section. However, these ideas are redeveloped by the authors in the Discussion section.
- General comment for “Methods”: Were the cells seeded first, thus several days (at least 24h) before treating them with SAHA or RA? If yes, please clarify this information.
- L114 – L116 (and title of Figure 2): If the cells were seeded on Petri dish covers, how these Petri dishes were then covered during the cell culture growth? Were the cells simply cultured in Petri dishes? What was the reason of tumor colonies’ (not droplets) transfer into 6-well plates? There is no risk to artificially modify the size of tumor spheroids?
- Figure 2C – D and the associated text: The size of tumor spheroids seems to be larger in RA treated cells than in SAHA treated ones. Is this difference statistically significant? If yes, please discuss the RA effect.
- Chapter 3.3: The authors should explain in the manuscript the type of staining corresponding to different death phenomena obtained with this annexin. Theoretically, the cells are stained as follows: (a) apoptotic cells bind annexin V (stained in green); (b) late apoptotic cells bind annexin V and PI (stained in yellow/orange); (c) necrotic cells bind PI (stained in red); (d) living cells have little or no fluorescence. Nevertheless, necrotic cells bind annexin V as well because phosphatidylserine is also exposed extracellularly. Therefore, it is difficult to distinguish between necrotic and late apoptotic cells. Could the authors explain what procedure they used to clearly discriminate between these two states? Was each cell state correctly identified in this experiment? The flow cytometry figures seem to be misinterpreted. If this is the case, the authors are invited to correct these data and rephrase their interpretation.
- Chapter 3.7, L279: Cleaved caspase-3 is the active form of this enzyme. The high caspase-3 expression is not necessarily associated with apoptosis, but the presence of cleaved form is indispensable. However, the cleaved form is not mentioned, neither in Methods (cf. Table 1) nor in Figure 7, where only total caspase-3 is shown. Therefore, the conclusions regarding cleaved caspase-3 are not supported by the reported data. This information needs to be clarified.
- Discussion (cell viability): The RA effects on normal (healthy) cells are they already known? Is it toxic for healthy cells? If no, these toxic effects on cancer cells, but not on healthy ones, can they be explained?
- Discussion, L324 (caspase-3). The information regarding caspase-3 must be clarified.
- L337 – L338: HDAC1 deacetylates p53 on its lysine residues (cf. ref. 40). HDAC does not acetylate histone on p53 (as written at L338). When not acetylated, these lysine residues are ubiquitinylated and p53 is proteolyzed by the proteasome (it is thus inactivated). Contrarily, acetylation positively regulates the p53 activity. Therefore, HDAC downregulates p53 activity, which cannot block anymore the cell cycle progression and trigger apoptosis. HDAC inhibition has the opposite effect and justifies the results reported in this manuscript. The authors are invited to clarify this information and the related discussion.
- L366: Figure 8 should be included in the main manuscript.
- Section 5, Conclusions: The last paragraph of “Discussions” could be transferred in the section “Conclusions”.
- Author Contributions: This section must be completed.
English syntax and grammar corrections
General observation: all abbreviations must be explained when first introduced in the text. The terms “in vivo” and “in vitro” must be formatted in italic font.
Abstract:
- L12: The sentence could be rephrased like this: “… has been known to present anti-inflammatory, anti-oxidant and anti-cancer effects”.
- L14: The sentence could be rephrased like this: “… gene expression by removing the acetyl group from histone.”
- L16: Please introduce comma after “HDAC2” and “8”. Then, the expression “has known to be“ may be replaced by “has been reported to be”.
- L17: “and downregulate” could be replaced by “where it downregulates”
- L22: “early and late stage apoptosis” instead of “apoptosis in an early and late stage”
- L24 – L25: The sentence could be rephrased like this: “PCNA, cyclin D1 and cyclin E1 were downregulated by RA, whereas p21 expression was upregulated”. The first part of the sentence (“As the results of HDAC2 inhibition by RA”) could be suppressed since this is a repetition of the previous sentence.
Introduction:
- L37: “Rosmarinus officinalis” must be formatted with italic fonts because this is a species name.
- L45: “Plus” could be replaced by “Moreover”.
- L49: Please replace “doing the enzymatic removal of acetyl group” by “removing the acetyl group”.
- L56: “phenomenon” should be in its plural from “phenomena”
- L56 – 57: “… epigenetic repression of tumor suppressor genes like CDKN1A (encoding the cyclin-dependent kinase inhibitor p21) and p53…” instead of “… epigenetic repression of the tumor suppressor gene like CDKN1A (encoding the cyclin-dependent kinase inhibitor p21), deacetylation of p53 …”
- L65: “the therapeutic properties of RA” instead of “treatment of RA”.
Methods:
- General comment: Were the reagents really ordered in the USA, or this location corresponds to the headquarter of different companies? If the reagents were ordered at a local distributor, its location should be mentioned.
- L80 and elsewhere in the manuscript: When you speak about two cell lines, you cannot designate them by the syntagma “two cells”. Please replace “two cells” by “two cell lines” in the entire manuscript.
- L83: What represents “cell maxin”?
- L85: “Both cell lines were detached by using 0.05% Trypsin-EDTA” instead of “The both cells were detached by using a 0.05% Trypsin-EDTA”
- L93: “The medium in 96-well plates was” instead of “All the medium in 96-well plates were”
- L94: “given” could be replaced by “dispensed”
- L99: “the ability of single cells to grow” instead of “the ability of single cells which grow”
- L99: “was replaced every 4 days” instead of “was replaced by every 4 days”
- L110 – L114: This sentence could be rephrased like this: “Hanging drop assay allows the formation of spheroid shaped tumors by self-assembly of tumor colonies, which can be used to evaluate chemotherapeutic drugs in a biological environment closer to in vivo models.”
- L120 – L125 (Annexin V assay): Several corrections are needed: (1) please suppress “apoptotic stages such as” because this is a redundant information; (2) the cell number should be expressed per volume; (3) the sentence at L124 should start with the adverb “Then” (“And” should be suppressed); (4) the binding buffer at L125 contains Alexa Fluor 488 conjugated annexin, not simply Alexa (which does not bind to apoptotic cells).
- L129 – L140 (TUNEL assay): Several corrections are needed: (1) to avoid redundant information, please suppress “with detection of the fluorescence of apoptotic cells” (L130); (2) “following the manufacturer instructions” instead of “instructions described in the kit” L132 – L133); (3) “After staining apoptotic cells” instead of “After apoptotic cells were stained”.
- L146 – L164 (Western blot analysis): Several corrections are needed: (1) “To determine the protein expression of HDAC2 and of genes involved in cell cycle and apoptosis regulation, western blot …” (L146 – L147); (2) “… for 48h. Then, whole cell …” (L148); (3) “which consists of” or “is composed of” instead of “is consisted of” (L149); (4) “(Sigma-Aldrich), 0.1% SDS and protease inhibitor” (L150); (5) “Protein mixtures (protein, distilled water (D.W) and dye) were then loaded …” (L152); (6) “Finally, the membranes were blocked …” (L155); (7) “Washing steps were performed by using 1x Tris …” (L157 – L159); (8) “Primary antibodies bound to membrane were detected with horse radish peroxidase (HRP)-conjugated anti-mouse IgG or anti-rabbit IgG (1:2000, Thermo Scientific, Waltham, MA, USA) incubated for 2 h at room temperature” (L159 – L161); (9) “The expression levels were quantified and normalized …” (L163).
- L166: “Statistical analysis” instead of “Data analysis”
- L167: “all data were statistically analyzed”
- L175: “DMSO was included in other medium as a vehicle” could be replaced by “DMSO was employed as a vehicle”
- L176: “the effect of HDAC inhibition, was assessed”
- L177: “RA was evaluated at”
- L181 – L183: “Based on these results, 1 μM SAHA and 200 μM RA were selected for further experiments.”
Results:
- Figures 1 – 7: “The results are expressed as means ± SD” instead of “All results are expressed as the means ± SD”
- L193: “After treatment with NC” instead of “After treatment of NC”
- L194 – L195: “The results showed a high number of colonies in NC, but only few colonies” instead of “The results showed that there were so many colonies in NC, but there were few colonies”
- L195 – L196: “According to these results, SAHA and RA” instead of “According to the results of colony formation assay, SAHA”
- L200 – L201: “The results showed that larger tumor spheroids were formed in NC, while small sized spheroids were detected in”
- L211: Please suppress the name of petri dishes supplier from the title of Figure 2.
- L214, title of chapter 3.3: “RA induced apoptosis in PCa cell lines” instead of “RA induced the apoptosis stage in PCa cell lines”
- L216: “treatment with NC, SAHA and RA”
- L221: Please suppress the adverb “also” from this sentence.
- Figure 3: Most of the text is illegible. Please increase the font size.
- Title of Figure 3: “apoptotic events” instead of “apoptotic activities”
- L232 – L233: “treatment with NC, SAHA and RA. Cell nuclei were stained in blue with DAPI, and DNA fragmentation occurring in apoptotic cells”
- L235: “apoptotic cells observed by DNA fragmentation”
- Figure 4: Most of the text is illegible. Please increase the font size.
- Title of Figure 4: “(A) Nuclei were stained with DAPI and DNA fragmentation …”
- Title of Figure 4: “DAPI and TUNEL were merged to observe apoptotic nuclei.”
- Title of Figure 4: “(B) The apoptotic cells presenting DNA fragmentation …”
- L246: “p53 expression at protein level”
- L248: “HDAC2 similarly to SAHA …”
- L250: “DU145 cells as compared to NC”
- L257: “p53 at protein level”
- L258: “were quantified and normalized to GADPH”
- L263: “Based on these results, RA significantly increased …”
- L271: “cyclin E1 at protein level”
- L272: “were quantified and normalized to GADPH”
- L275: “The protein expression of genes related with apoptosis was confirmed”
- L278: “treatment with SAHA …”
- L286: “PARP-1 (cleaved) at protein level”
- L287: “were quantified and normalized to GADPH”
- L296 – L297: “However, its anticancer effect and related mechanisms on PCa have not yet been known”
- L321 – L322: “both PCa cell lines, which was identified by TUNEL assay”
- L350: “anti-cancer effects” instead of “anti-cancer efficacies”
- L352: “achieved through” instead of “achieved by through”

Author Response1

Round 2
Reviewer 1 Report
This revised paper is NOT significantly improved. The use of supraphysiological dose of rosmarinic acid in this study is not well justified. The response to reviews’ comments revealed that the authors lack basic knowledge of TP53 biology, which is a very important biomarker in this study. The authors claimed that "We focused on the DO-1 form of p53". However, there is no "DO-1 form of p53". Furthermore, this antibody is not specified for which isoform of p53 is recognized.
Author Response
This revised paper is NOT significantly improved. The use of supraphysiological dose of rosmarinic acid in this study is not well justified. The response to reviews’ comments revealed that the authors lack basic knowledge of TP53 biology, which is a very important biomarker in this study. The authors claimed that "We focused on the DO-1 form of p53". However, there is no "DO-1 form of p53". Furthermore, this antibody is not specified for which isoform of p53 is recognized.
Ă For doses of RA: As pointed out by the reviewer again, supraphysiologic doses of RA to produce anticancer effects on PCa cell lines were used in the present study. As replied to the former comment, the supraphysiologic doses of RA such as 200 uM are not decisive doses for clinical applications because further studies for identifying the effective in vivo doses of RA are definitely required. In case of need for supraphysiologic doses of RA in vivo, it is considered possible to reach high levels of RA by supplying it as a dietary additive or a drug containing its effective therapeutic concentration. Therefore, we may find the value of this study in the demonstration of the possibility of RA as a natural phytochemical for antiprostate cancer therapy.
Ă For TP53 biology: We are sorry to not fully understand your question “which isoform of p53 was detected?” In the present study, we used a DO-1 form antibody of p53 (p53 (DO-1): sc-126) supplied by SANTA CRUZ BIOTECHNOLOGY, INC. DO-1 is a mouse monoclonal antibody only specific for p53 (FLp53), p53β and p53Îł (Cold Spring Harb Perspect Biol., 2010; 2(3): a000927, PMID: 20300206). FLp53 as a canonical p53 protein is the first p53 isoform to be identified and also named p53, FLp53, p53α or TAp53α. Also, p53β and p53Îł are other isoforms of p53 with spilced oligomerization domain (OD) at C-terminal. According to a review article (Uncovering the role of p53 splice variants in human malignancy: a clinical perspective. OncoTargets and Therapy 2014:7 57–68), canonical p53 is well known as a tightly regulated major tumor suppressor. p53β can enhance p53 transcriptional activity on the p21 and Bax promoters. It can also induce apoptosis in a p53-independent manÂner. For p53Îł, it can enhance p53 transcriptional activity on the Bax promoter, but not on the p21 promoter. p53Îł can also regulate gene expression independently of p53 and has cytotoxic activity. Therefore, it is considered to be appropriate to use DO-1 form antibody of p53 to verify the roles of RA in induction of cell cycle arrest and apoptosis of PCa cells.
1. Rosmarinic acid has anticancer properties against different cancer types, such as gastric cancer, colorectal cancer, liver cancer, breast cancer, ovarian cancer, etc. This information should be included in Introduction section.
Ă The information about anticancer effects of Rosmarinic acid (RA) on diverse cancers has been added in Introduction section, as requested.
2. In this in vitro study, the dose of rosmarinic acid starts from 25 µmoml/L and the highest concentration is 300 µmol/L. The concentration to achieve >50% cell viability inhibition is more than 200 µmol/L.
In animal study: The level of plasma total rosmarinic acid (free and conjugate forms) reached a maximum concentration of 4.63µmol/L after orally administration to the rats at a dose of 50mg/kg body weight (Life Sci. 2004, 75:165-178, PMID 15120569).
In human study: The level of serum total rosmarinic acid reached a maximum concentration of 162.20 nmol/L after orally administration to healthy individuals at a dose of 500mg rosmarinic acid (PLoS One. 2015, 10:e0126422, PMID: 25978046).
Based on the literature report, it seems that the authors used a supraphysiological dose of rosmarinic acid to achieve favorable effects. How could the findings in this study be translated into clinical practice?
à As pointed out by the reviewer, the concentration of RA tested in this study (200 μM) was a supraphysiological level. In order to identify the anti-prostate cancer effect and the underlying mechanism of RA in vitro, we used the high dose of RA, as in other studies (200 μM RA used in “Rosmarinic Acid Activates AMPK to Inhibit Metastasis of Colorectal Cancer”. Front Pharmac ol 9:68. 2018/40 μM & 160 μM RA used in “Anticancer effects of Rosmarinic acid in OVCAR-3 ovarian cancer cells are mediated via induction of apoptosis, suppression of cell migration and modulation of lncRNA MALAT-1 expression”. J BUON. 2018 May-Jun;23(3):763-768). In fact, it is not possible to translate the findings of this study into clinical practice because we identified antiprostate cancer effect of RA in the in vitro cellular model only, not in the in vivo animal model. Therefore, we should determine the effective in vivo dose of RA in further studies. For instance, in case that higher concentrations of RA than its physiological concentration are needed to target prostate cancer, it seems to be necessary to develop RA into anti-prostate cancer medicine and use it actively for medical purposes, not only depending on its dietary ingestion from foods including RA. In conclusion, it can be said that this study provides a possibility of RA for antiprostate cancer therapy and needs more follow-up researches.
3. The rosmarinic acid concentration should be indicated in tumor spheroid formation assay, Annexin V assay, and TUNEL assay (page 4).
Ă The concentrations of SAHA and RA used in those assays have been indicated, as requested.
4. There are at least 12 isoforms of p53 proteins. Some p53 proteins act as tumor suppressor, and some of which have tumor-promoting properties. Which isoforms of p53 protein was detected in this study?
Ă We focused on the DO-1 form of p53, which is known as a tumor suppressor to promote growth arrest and apoptosis, and purchased its antibody [sc-126 from Santa Cruz Biotechnology] to analyze its protein expression in western blot assay.

Reviewer 2 Report
The Authors have accomplished all my suggestions improving their manuscript
Reviewer 3 Report
This Reviewer is satisfied by the revised manuscript, which was extensively amended according to the Reviewers’ suggestions. However, a few points need further clarification before the final decision, as explained below:
- Chapter 2.6: For clarity, please add the abbreviation “(Alexa)” just after the complete name of the compound “Alexa Fluor 488 annexin V”. Then, suppress “Alexa Fluor 488” and the parentheses around “Alexa” at L219.
- Chapter 3.1 (L296): Please rephrase the sentence: either “SAHA and RA were administered to both cell lines” or “Both cell lines were treated with SAHA and RA”.
- Chapter 3.1 (L297): Please explain the abbreviation “WST”.
- Chapter 3.3 (L367 – 368): The statement that “Annexin V binds apoptotic or dead cells” needs to be corrected. Apoptotic cells are also dead by a programmed cell death mechanism. Moreover, cell death can be triggered by various mechanisms: necrosis, apoptosis, necroptosis, pyroptosis etc. Therefore, it may be appropriate to state that “Annexin V detects early and late stages of apoptosis”. In this context, the authors could explain within the manuscript what combination of Annexin V and PI corresponds to each cell state, i.e. early apoptosis, late apoptosis, necrosis, live state.
- Chapter 3.7 (L464 – L465): Please introduce the verb “was” between “Caspase-3” and “upregulated”. Please rephrase “was cleaved in the cells treated with RA only” (or as you prefer). This is because PPARP-1 was not cleaved by treatment with RA, but by caspase-3.
- Figures 3 and 4: There must be a misunderstanding. The text within the figures themselves is illegible not that in the Figure captions. The font size within figures depends on formatting done by authors and not on the Editor’s decision. The authors are thus invited to enlarge the font size of some legends in these figures, which cannot be read.
- In Discussion or Conclusions, the authors could explain that further studies are necessary to identify eventual cytotoxic effects of RA on normal cells. They could also add that supraphysiologic doses of RA are required to produce anti-cancer effects, which could be eventually attained by regular consumption of RA as a dietary additive or via the formulation of a drug containing the effective therapeutic concentration.
Author Response
This Reviewer is satisfied by the revised manuscript, which was extensively amended according to the Reviewers’ suggestions. However, a few points need further clarification before the final decision, as explained below:
- Chapter 2.6: For clarity, please add the abbreviation “(Alexa)” just after the complete name of the compound “Alexa Fluor 488 annexin V”. Then, suppress “Alexa Fluor 488” and the parentheses around “Alexa” at L219.
à The abbreviation “Alexa” has been described as a full name “Alex Fluor 488 annexin V (Alexa)” and then, abbreviation “Alexa” has been used as requested.
- Chapter 3.1 (L296): Please rephrase the sentence: either “SAHA and RA were administered to both cell lines” or “Both cell lines were treated with SAHA and RA”.
Ă The sentence has been corrected as requested.
- Chapter 3.1 (L297): Please explain the abbreviation “WST”.
Ă The full name of WST, Water Soluble Tetrazolium salt, assay has been added.
- Chapter 3.3 (L367 – 368): The statement that “Annexin V binds apoptotic or dead cells” needs to be corrected. Apoptotic cells are also dead by a programmed cell death mechanism. Moreover, cell death can be triggered by various mechanisms: necrosis, apoptosis, necroptosis, pyroptosis etc. Therefore, it may be appropriate to state that “Annexin V detects early and late stages of apoptosis”. In this context, the authors could explain within the manuscript what combination of Annexin V and PI corresponds to each cell state, i.e. early apoptosis, late apoptosis, necrosis, live state.
Ă The sentence has been corrected and the combination of Annexin V and PI in each state has been indicated - necrosis (Annexin V-negative/PI-positive), late apoptosis (Annexin V-positive/PI-positive), live (Annexin V-negative/PI-negative), early apoptosis (Annexin V-positive/PI-negative) as requested.
- Chapter 3.7 (L464 – L465): Please introduce the verb “was” between “Caspase-3” and “upregulated”. Please rephrase “was cleaved in the cells treated with RA only” (or as you prefer). This is because PPARP-1 was not cleaved by treatment with RA, but by caspase-3.
à The verb “was” has been added and that sentence has been corrected as requested.
- Figures 3 and 4: There must be a misunderstanding. The text within the figures themselves is illegible not that in the Figure captions. The font size within figures depends on formatting done by authors and not on the Editor’s decision. The authors are thus invited to enlarge the font size of some legends in these figures, which cannot be read.
Ă The font size of Figure 3 and 4 has been enlarged and it will be sent to editor.
- In Discussion or Conclusions, the authors could explain that further studies are necessary to identify eventual cytotoxic effects of RA on normal cells. They could also add that supraphysiologic doses of RA are required to produce anti-cancer effects, which could be eventually attained by regular consumption of RA as a dietary additive or via the formulation of a drug containing the effective therapeutic concentration.
Ă The additional reqiurements of the reviewer were reflected in the Conclusions as follows.
5. Conclusions
RA, as a dietary phenolic compound ingested from tea, was displayed to have anti-prostate cancer activities by inhibiting viability, colony formation, and spheroid formation of PCa cells via HDAC2 inhibition and the consequential p53-mediated cell cycle arrest and apoptosis. Therefore, RA would be used as a novel phytomedicine to act as a HDAC inhibitor targeted to prostate cancer with the anticipation to decrease the adverse side effects of the existing chemotherapeutical agents. To do this, further studies are necessary to identify eventual cytotoxic effects of RA on normal cells. In addition, supraphysiologic doses of RA should be attained to produce anti-cancer effects on PCa via the formulation containing RA as a dietary additive or a drug containing its effective therapeutic concentration.
